# Altered auditory feature discrimination in a rat model of Fragile X Syndrome

D. Walker Gauthier[1,2,3], Noelle James[1,2], Benjamin D. Auerbach[1,2,3]*

1 Department of Molecular & Integrative Physiology, School of Molecular & Cellular Biology, University of Illinois Urbana-Champaign, Urbana, Illinois, United States of America, 2 Beckman Institute for Advanced Science & Technology, University of Illinois Urbana-Champaign, Urbana, Illinois, United States of America, 3 Neuroscience Program, University of Illinois Urbana-Champaign, Urbana, Illinois, United States of America

* bda5@illinois.edu

## Abstract

Atypical sensory processing, particularly in the auditory domain, is one of the most common and quality-of-life affecting symptoms seen in autism spectrum disorders (ASD). Fragile X Syndrome (FXS) is a leading inherited cause of ASD and a majority of FXS individuals present with auditory processing alterations. While auditory hypersensitivity is a common phenotype observed in FXS and *Fmr1* knockout (KO) rodent models, it is important to consider other auditory coding impairments that could contribute to sound processing difficulties and disrupted language comprehension in FXS. We have shown previously that a *Fmr1* KO rat model of FXS exhibits heightened sound sensitivity that coincided with abnormal perceptual integration of stimulus bandwidth, indicative of altered spectral processing. Frequency discrimination is a fundamental aspect of sound encoding that is important for a range of auditory processes, such as source segregation and speech comprehension, and disrupted frequency coding could thus contribute to a range of auditory issues in FXS and ASD. Here we explicitly characterized spectral processing deficits in male *Fmr1* KO rats using an operant conditioning tone discrimination assay and in vivo electrophysiological recordings from the auditory cortex and inferior colliculus. We found that *Fmr1* KO rats exhibited poorer frequency resolution, which corresponded with neuronal hyperactivity and broader frequency tuning in auditory cortical but not collicular neurons. Using an experimentally informed population model, we show that these cortical physiological differences can recapitulate the observed behavior discrimination deficits, with decoder performance being tightly linked to differences in cortical tuning width and signal-to-noise ratios. Together, these findings indicate that cortical hyperexcitability in *Fmr1* KO rats may act to preserve signal-to-noise ratios and signal detection threshold at the expense of sound sensitivity and fine feature discrimination, highlighting a potential mechanistic locus for a range of auditory behavioral phenotypes in FXS.

**Data availability statement:** Pre-processed data and analysis code are publicly available at https://zenodo.org/records/15559344.

**Funding:** This work was supported by the National Institutes of Health (NIH) National Institute on Deafness and Other Communication Disorders (NIDCD, https://www.nidcd.nih.gov/) award K01DC018310 to BDA, and Eunice Kennedy Shriver National Institute of Child Health and Human Development (NICHD, https://www.nichd.nih.gov/) award R01HD111753 to BDA. The funders had no role in study design, data collection and analysis, decision to publish, or preparation of the manuscript.

**Competing interests:** The authors have declared that no competing interests exist.

**Abbreviations:** ASD, Autism Spectrum Disorders; FXS, Fragile X Syndrome; RT, reaction time; FA, false alarm; CR, correct rejection; FRA, frequency response area; RLF, rate level function; ABR, auditory brainstem recording; IC, inferior colliculus; ACx, auditory cortex; CF, characteristic frequency; I/O, input/ouput.

## Introduction

Fragile X syndrome (FXS) is a leading monogenetic cause of intellectual disability and autism spectrum disorders (ASD), resulting from the transcriptional silencing of the FMR1 gene and subsequent loss or reduction in its protein product, fragile x messenger ribonucleoprotein (FMRP) [1]. Sensory processing difficulties are a defining feature of FXS, contributing to sensory avoidance, anxiety, altered communication, and disrupted social behavior [2,3]. A common theme across sensory modalities in FXS is hypersensitivity, with many lines of evidence pointing to increased evoked response magnitude and/or impaired habituation to repetitive stimuli in FXS individuals and rodent models [4–10]. Sensory issues in FXS and ASD, however, are not limited to changes in perceptual sensitivity, with reported deficits in visual discrimination [11–13], delays in speech comprehension and language development [14–16], and difficulties navigating complex or cluttered sensory environments [17,18]. Likewise, sensory coding impairments in FXS rodent models manifest in many ways beyond changes to evoked response size, such as elevated spontaneous activity [19], altered network synchronization [19–21], and broadened tuning or blurred sensory maps [11,22–25]. Linking these physiological changes to their perceptual consequences is crucial for understanding the relationship between sensory processing alterations in *Fmr1* knockout (KO) models and the perceptual abnormalities reported in FXS individuals.

Altered sound processing is a particularly prominent and quality-of-life affecting sensory issue in FXS [2]. We previously observed that a *Fmr1* KO rat model of FXS exhibits behavioral evidence for auditory hypersensitivity in the form of faster reaction times in a Go/No-go sound detection task [26]. In addition to this heightened sensitivity, we found that male *Fmr1* KO animals exhibited abnormal perceptual integration of stimulus bandwidth, suggesting that spectral processing differences may be present as well. The ability to discriminate sound frequency is a fundamental feature of the auditory system important for source segregation and speech comprehension, and disrupted frequency coding could thus contribute to a range of auditory issues in FXS and ASD [27–29]. Broadened frequency tuning has indeed been observed in the auditory cortex of *Fmr1* KO mice [23], which could impact feature discrimination by blurring differences in population activity across different stimuli [30]. However, the perceptual consequences of these tuning differences on sound discrimination behavior remain unknown.

In this study, we demonstrate that male *Fmr1* KO rats exhibit impaired fine-frequency discrimination in an operant Go/No-go tone discrimination task, despite normal learning and detection thresholds. Parallel in vivo electrophysiological recordings from the auditory cortex and midbrain demonstrated that these discrimination deficits coincided with elevated spontaneous rates, increased sound-evoked activity, and broader frequency tuning in the auditory cortex of *Fmr1* KO rats, while subcortical response properties were largely unaffected. By using an experimentally informed population decoder, we show that these cortical physiological differences can recapitulate the observed behavior discrimination deficits, with decoder performance being

tightly linked to differences in tuning width and signal-to-noise ratios. These results indicate that fine feature discrimination deficits in *Fmr1* KO rats are associated with cortical hyperexcitability and degraded frequency tuning, providing insight into the nature of sensory processing difficulties in FXS and their underlying neural mechanisms.

## Materials and methods

### Ethics statement

All experiments were approved by the University of Illinois Urbana-Champaign Institutional Animal Care and Use Committee (IACUC) under protocol 20252, in accordance with NIH guidelines.

### Subjects

Adult (>3 month old) male *Fmr1^tm1sage^* knockout rats on an outbred Sprague-Dawley background (TGRS5390HTM4 FMR1 -/Y; SAGE Labs, St. Louis, MO) and littermate wild-type controls were used for these studies. Littermate male wild-type (WT) and *Fmr1^-/y^* (KO) rats were generously donated from the laboratory of Dr. Richard Salvi. Male rats were used because FXS occurs more frequently and in greater severity in males due to the X-linked nature of the disorder [31]. Offspring were screened for a 122-base pair (bp) deletion in the *Fmr1* gene sequence using published procedures [26,32].

9 WT and 10 *Fmr1* KO rats were used for operant conditioning experiments. 6 WT and 5 *Fmr1* KO rats were used for auditory brainstem response recordings. 5 WT and 5 *Fmr1* KO rats were used for extracellular depth recordings. Rats lived in pair-housed caging in a colony room maintained at 22 °C with a 12-h light–dark cycle. All subjects had free access to food and water except for those undergoing operant conditioning, when rats were food restricted and kept at approximately 90% of their free-feeding weight. Food-restricted animals had unrestricted access to water, except while participating in behavioral testing. Testing sessions lasted approximately 1 h per day, with rats participating in one behavioral testing session per day, 6 days per week.

### Operant conditioning

Rats were first trained in a *Go/No-Go* operant conditioning paradigm to detect tone bursts (300 ms, 5 ms rise/fall time, cosine gated) of varying frequency (4, 8, 16, 32 kHz) using procedures similar to those described in our previous publication [26]. Testing was carried out in operant conditioning chambers (Med-Associates, Model ENV-008-VP, St. Albans, VT) equipped with pellet dispensers (Med-Associates, Model ENV-203 M, St. Albans, VT), illuminated nose-pokes with infrared sensor (Med-Associates, Model ENV-114BM, St. Albans, VT), cue lights for signaling task status (Med-Associates, Model ENV-215M-LED, St. Albans, VT), and speakers (Fostex F17H Horn Super Tweeter, Tokyo, Japan) for sound generation. All components were housed in single-walled sound-attenuating cubicles equipped with ventilation (Med-Associates, Model ENV-018MD, St. Albans, VT). Each behavioral setup was independently controlled by a dedicated microcontroller (Arduino DUE, Italy) that recorded signals from the behavioral devices, responded according to predefined protocols, and communicated with a host computer to transmit control signals and data. The entire system was controlled by custom software written in MATLAB (MathWorks, R2022, USA), which handled system control, stimulus generation, and data collection and visualization. Sound stimuli (192 kHz sampling rate) were generated and delivered using a multi-channel sound card (RME MADIface Pro, Germany), multi-channel digital-to-analog converter (RME M-32 DA Pro, Germany), power amplifiers (Behringer NX1000, Germany), and speakers (Fostex F17H Horn Super Tweeter, Tokyo, Japan) positioned approximately 30 cm above the animal's head. Stimuli were calibrated using a microphone preamplifier (Larson Davis, Model 2221, Depew, NY) equipped with a ½″ microphone (Larson Davis, Model 2520, Depew, NY) at a location where the animal's head would be during a trial.

A rat began a trial by placing its nose in the nose-poke hole, which initiated a variable waiting interval ranging from 1 to 4 s. During the waiting interval, the rat had to maintain its position in the nose-poke hole until it heard a tone burst or the trial was aborted. In the *Go* condition, the target stimulus was the tone burst. If the rat detected this signal, it removed its

nose from the nose-poke hole resulting in a food reward (45 mg dustless rodent grain pellets, Bio-Serv, Flemington, NJ); a *hit* was recorded if the rat correctly responded to the tone within 2 s. A *miss* was recorded if the rat failed to remove its nose from the nose-poke within the 2 s response interval. No reinforcement was given for a *miss*. Approximately 30% of all trials were *catch* trials where tone bursts were not presented. This constituted the *No-Go* part of the procedure. If the rat removed its nose during a *catch* trial, a *false alarm* (FA) was recorded and the rat received a 4–8 s timeout, during which the house light was turned off and the rat could not start another trial. However, if the rat continued to nose-poke, a *correct rejection* (CR) was recorded. No reinforcement was given for a *correct rejection*. After initial training using a 70 dB SPL *Go* stimulus (training criteria: >240 trials, >90% hit rate, <25% FA rate over 5 consecutive days), the range of stimulus intensities was expanded to 30–90 dB SPL, presented in 10-dB steps, and then to 15–45 dB, presented in 5 dB steps, to obtain a detection threshold for each frequency. If necessary, the sound level range was further reduced in 5 dB steps until thresholds were obtained. The tone bursts were presented according to the psychophysical Method of Constant Stimuli. Within each 10-trial block, seven predetermined target intensities were presented randomly along with 3 *catch* trials without sound stimulus.

Following collection of tone detection thresholds, the rats were moved to the discrimination phase of the *Go*/*No-Go* task. In this phase, the rats were trained to respond to a single *Go* tone frequency (4, 8, 16, or 32 kHz) while the *No-Go* stimulus was a tone either 1 octave above or below the *Go* tone. Yoked pairs of WT and *Fmr1* KO rats were trained on one of six potential *Go*/*No-Go* tone pairings (4/8 kHz, 8/4 kHz, 8/16 kHz, 16/8 kHz, 16/32 kHz, 32/16 kHz). Both the *Go* and *No-Go* stimulus were presented at 40 dB above sensation level (SL), based on the individual subject's detection thresholds acquired above. If the rat responded to the *No-Go* tone by removing its nose from the nose-poke hole within a 2 s response window, this was recorded as a FA and the rat received a 4–8 s time-out. The percentage of *No-Go* trials presented was incrementally increased over each session as the animals improved on the discrimination task until sessions were 50:50 Go/No-Go trials. Once rats reached criteria (>200 trials, >90% hit rate, <25% FA rate over 5 consecutive days), they could begin the discrimination testing phase. In the first portion of this testing phase, 5 new *No-Go* tones were introduced to the *Go*/*No-Go* task for 2 consecutive sessions. These tones were equally spaced at 1/6th octave steps between the *Go* tone and original *No-Go* tone (e.g., Go: 8 kHz, No-go: 8.9, 10.1, 11.3, 12.7, 14.3, 16 kHz). Rats were then tested in a more fine-grained version of the task using 12 *No-Go* tones equally spaced in 1/12th octave steps above or below the *Go* tone. This fine feature phase was split over three sessions, where the rats were tested on *No-go* tones >2/3 octave from *Go* (e.g., *Go*: 8 kHz, *No-go*: 13.5, 14.3, 15.1, 16 kHz), 1/3–2/3 octave from *Go* (e.g., *Go*: 8 kHz, *No-go*: 10.7, 11.3, 11.9, 12.7 kHz), or <1/3 octave from *Go* (e.g., *Go*: 8 kHz, *No-go*: 8.5, 8.9, 9.5, 10.1 kHz) on any given session in pseudorandom order. Fine-frequency testing sessions were interleaved with 1 octave *Go*/*No-Go* training sessions to ensure consistent reinforcement. For all discrimination testing sessions, trials were 50:50 *Go*/*No-Go* and if the subject responded to any of the *No-Go* tones within the 2 s response window, a FA was recorded and the animal received a 4–8 s timeout.

### Auditory Brainstem Response (ABR) recordings

Animals were anesthetized with a cocktail containing ketamine (80 mg/kg i.p.) and xylazine (6 mg/kg i.p.). Platinum sub-dermal needle electrodes (S83018-R9-10, Horizon) were placed behind each ear to record auditory brainstem response (ABR), with ground and reference electrodes placed on the rear of the body and the vertex of the head, respectively. Body temperature was maintained at 37 °C using a homoeothermic heating blanket (Harvard Apparatus, Cambridge, MA). Tone bursts (5 ms, 1 ms rise/fall time, cosine gated) were generated by a RX6-2 multifunction processor (Tucker-Davis Technology, Alachua, FL) at approximately 250 kHz sampling rate and presented at a rate of 21/s. The output of the D/A converter was routed through a programmable attenuator (PA5, Tucker-Davis Technology, Alachua, FL) and amplifier (SA1, Tucker-Davis Technology, Alachua, FL), then delivered through a loudspeaker (MF1, Tucker-Davis Technology, Alachua, FL) located 10 cm from the test ear, while the other ear was plugged. The stimulation intensity varied from 0 to 90 dB SPL

with 10-dB steps from 30-90 and 5 dB steps from 0 to 25 dB. Sound levels were calibrated with a ½″ microphone (Larson Davis, Model 2540, Depew, NY), a microphone preamplifier (Larson Davis, Model 2221, Depew, NY) and custom sound calibration software written in MATLAB (MathWorks, R2022, USA). The signals from the electrodes were amplified and digitized using a Medusa4Z amplifier and RZ2 Bioamp processor (Tucker-Davis Technology, Alachua, FL). All experiments were conducted in a double-walled sound-insulated chamber. Signals were band-pass filtered (300–3000 Hz) and averaged over 500 repeats using custom written MATLAB script that allows for the visual inspection of average ABR waves at each intensity-frequency combination. ABR threshold was defined as the lowest level that produced a noticeable ABR waveform, determined by visual inspection by a trained experimenter blind to genotype. Thresholds were determined for both ears and then averaged within each animal, except for two WT rats in which measurements were limited to a single ear due to insufficient anesthesia depth that precluded bilateral testing.

**Extracellular depth recordings**

Extracellular recordings of spontaneous and tone-evoked neural activity were performed in anesthetized animals as described in previous publications [33,34]. Rats were anesthetized with a cocktail of ketamine (80 mg/kg, i.p.) and xylazine (6 mg/kg, i.p.) and placed in a stereotaxic frame with blunted ear bars (Kopf Instruments, Tujunga, CA). Supplementary doses of ketamine/xylazine (20/2 mg/kg, i.p.) were administered as needed to maintain a stable plane of anesthesia. Body temperature was maintained at 37 °C using a homoeothermic heating blanket (Harvard Apparatus, Cambridge, MA). The dorsal surface of the skull was exposed, the temporal muscle was removed, and craniotomies were made over the left or right (contralateral to the sound source) auditory cortex (ACx) and inferior colliculus (IC) based on stereotaxic coordinates [35]. The skull was carefully opened and the dura was removed from the surface of the cortex. A custom designed head-bar was firmly attached to the skull using dental cement to allow for acoustic stimulation using a free-field loudspeaker.

Two single-shank 16-channel linear silicon microelectrode arrays (A-1 × 16–10 mm 100–177, NeuroNexus Technologies, Ann Arbor, Michigan) were inserted tangential to the surface of the ACx (approximately 40°) and IC (90°) to record multi-unit spiking activity. Recordings were made at several different positions along the rostral-caudal axis of each area (ACx: 4.0–6.0 mm caudal from Bregma in 0.5 mm steps, approximately 7.6 mm lateral from midline; IC: 8.4–9.6 mm caudal from Bregma in 0.4 mm steps, approximately 1.8 mm lateral from midline). Electrodes were slowly advanced using a hydraulic micromanipulator (FHC, Bowdoinham, ME) to a depth of approximately 1.6 mm in the ACx or approximately 3 mm in the IC so that for each penetration, the electrode shank spanned layers of the ACx and iso-frequency lamina of the IC (see Fig 3A). Recordings were targeted to primary ACx and the central nucleus of the IC based on stereotaxic coordinates, response latency, and tuning properties. For each penetration, electrodes were allowed to settle for at least 30 min prior to data collection.

To examine rate level functions (RLFs) and frequency response areas (FRAs), tone bursts (50 ms, 1 ms rise/fall time, cosine gated) were generated by a TDT RX6-2 multifunction processor (approximately 250 kHz sampling rate) and delivered at a rate of 3/s through a loudspeaker (MF1, Tucker-Davis Technologies, Alachua, FL) located 10 cm from the ear contralateral to the recording hemisphere. Sound levels at the position of the ear were calibrated using a microphone preamplifier (Larson Davis, Model 2221, Depew, NY) equipped with a ½″ microphone (Larson Davis, Model 2520, Depew, NY). Tone stimuli spanned from 1 to 64 kHz in 20 log steps, presented from 0 to 90 dB SPL in 10 dB steps. Thirty repeats of each frequency–intensity combination were presented in pseudo-random order. Recorded neural signals were amplified by a RA16PA and sampled at 25 kHz by a RZ5 Bioamp processor (Tucker-Davis Technologies System-3, Alachua, FL). Spike detection was performed online using a manually set voltage threshold (spike signal filtered 300–3,500 Hz). Custom-written MATLAB software (MathWorks, R2022, USA) was used to acquire the neural data as previously described [33,36]. All experiments were conducted in a double-walled sound-insulated chamber.

PLOS Biology

## Data analysis

Multi-unit spiking activity recorded in response to tone bursts was analyzed using a custom-built Python (3.10) pipeline. Spiking data was extracted into a tuning matrix (dB × kHz) for each multi-unit, with each cell corresponding to 30 spike trains from −40 ms pre-stimulus to 100 ms post-stimulus for each presentation of the current dB/kHz combination. Peristimulus time histograms (PTSHs) were then calculated for each cell using 5 ms bins. Cells were determined to have evoked spiking activity if one 5 ms bins from 0 to 40 ms post-stimulus exceed 4 standard deviations above the mean pre-stimulus activity [37].

**Rate level functions (RLFs).** After data extraction, RLFs were produced for each frequency column of the dB × kHz matrix. Frequency specific input/output (I/O) functions were calculated as firing rate change over increasing sound intensity for a given frequency (4, 8, 16, and 30 kHz). In order to compare across units with different tuning properties, I/O functions were constructed for each multi-unit cluster using only the frequency column corresponding to the unit's characteristic frequency (CF) (see CF methods below). This I/O function was then fit with the following 6 parameter Gaussian model [38]:

$$y = a + \frac{d-a}{1 + e^{\frac{b-x}{c}}} + e \cdot e^{-\frac{(x-f)^2}{2c^2}} \tag{1}$$

Where $a$ is the lower asymptote, $d$ is the upper asymptote, $b$ is the inflection point, $c$ is the variance, $e$ is the amplitude, and $f$ is the center of the Gaussian. Response minimum and maximum were defined as the parameters $a$ and $d$, respectively. The threshold was calculated as 20% of the range from the minimum asymptote to the maximum [38]. Gain was taken as the slope of the ascending portion of the RLF [38]. A root mean squared error (RSME) cutoff of <100 was used to limit analysis only to RLFs adequately fit by the model (233/336 ACx units and 166/192 IC units from WT animals; 208/272 ACx units and 153/176 IC units from *Fmr1* KO animals)

**Frequency response areas (FRAs).** FRAs were analyzed by constructing PSTHs for each cell of the tuning matrix (dB × kHz), as described above. Characteristic frequency (CF) was defined as the frequency value which had the lowest threshold (minimum threshold). Threshold was determined as the lowest intensity to produce a sound-evoked response (>4 SD above mean pre-stimulus firing rate). To characterize tuning quality, bandwidth was determined for responses 10, 20, 30, and 40 dB above threshold, defined as the first and last cells for each intensity row to produce an evoked response. This bandwidth was then used to calculate $Q$-values at each of these intensities by dividing the CF by the bandwidth, accounting for tuning differences due to CF location. Units were manually curated by a trained experimenter blind to genotype to only include those with tone-evoked responses and discernible tuning curve (224/336 ACx units and 170/192 IC units from WT animals; 226/272 ACx units and 158/176 IC units from *Fmr1* KO animals).

**Spike train discriminability.** In order to determine how well a multi-unit cluster could discriminate between different frequency tones, neural discriminability was quantified using SPIKE-distance, which calculates the dissimilarity between spike train pairs based on differences in spike timing and instantaneous firing rate [39]. For a given spike train, the instantaneous spike timing difference at time $t$ is given as:

$$S_1(t) = \frac{\Delta t_P^{(1)}(t) x_F^{(1)} + \Delta t_F^{(1)}(t) x_P^{(1)}}{x_{ISI}^{(1)}(t)}, t_P^{(1)} \leq t \leq t_F^{(1)} \tag{2}$$

where $\Delta t_P$ corresponds to the distance between the spike before time $t$ in spike train 1 and the closest spike from train 2 and $\Delta t_F$ corresponds to the same distance, but for the spike following time $t$. $S_2(t)$ takes the same form, but from the view of the second spike train in the pair with respect to the first. The two train contributions are then averaged and normalized by the mean to account for relative distances within spike trains. Because mean firing rates were significantly different between WT and *Fmr1* KO animals, rate independent SPIKE-distance was used. In this manner, $S_1(t)$ and $S_2(t)$ are left unweighted by the instantaneous ISI:

$$S'(t) = \frac{S_1(t) + S_2(t)}{2 \left\langle x_{ISI}^{(n)}(t) \right\rangle_n^2} \tag{3}$$

Distance calculations were performed by comparing the block of 30 spike trains at a unit's CF to the spike trains from adjacent frequencies. All comparisons were made for tone-evoked spike trains at 40 dB above minimum threshold (corresponding to the intensity used for behavioral discrimination experiments). In this manner, a 30 × 30 matrix for every pairwise distance frequency combination relative to CF was produced. Due to the large number of spike trains being compared ($N > 2$), the instantaneous average over all pair combinations of instantaneous spiking disparities was taken:

$$S^a(t) = \frac{1}{N(N-1)/2} \sum_{n=1}^{N-1} \sum_{m=n+1}^{N} S^{mn}(t) \tag{4}$$

Lastly, these instantaneous averages were then integrated across time:

$$D_S^a = \frac{1}{T} \int_{t=0}^{T} S^a(t) dt \tag{5}$$

## Population decoder

To characterize the relationship between neural tuning and behavioral discrimination, we constructed a Bayesian population decoder consisting of a population of readout neurons that received frequency tuned input from either ACx or IC recorded neural data [40]. First, we determined the response maximum, minimum, and bandwidth for each multi-unit cluster at 40 dB above threshold (corresponding to the intensity used in our behavioral discrimination experiments) from the ACx or IC recordings to fit asymptotic Gaussians models of the firing data using the following equation:

$$f_{(x)} = A \cdot exp\left(-\frac{(x-B)}{2\sigma^2}\right) + N \tag{6}$$

Where $A$ is the amplitude (peak firing rate), $x$ is in the input variable (tone frequency), $B$ is the mean center of the Gaussian (best frequency), $\sigma$ indicates the variance (tuning width), and $N$ is the asymptote (baseline firing rate). Given that the tuning edges were defined as the first and last frequencies that exhibit an evoked response (regardless of the peak response size), the bandwidth value informing the Gaussian width would be equivalent to the first divergence of the Gaussian from the horizontal asymptote. This cannot be analytically solved for but can be approximated as the 99% confidence interval expressing the cumulative density function in terms of the error function (*Erf*):

$$P(X \le x) = \frac{1}{2}\left[1 + Erf\left(\frac{x-\mu}{\sigma\sqrt{2}}\right)\right] \tag{7}$$

We can then solve for the corresponding sigma by using 99% as the $P(X \le x)$.

Next, two populations of 640 neurons were generated with preferred frequencies equally log-spaced from 1 to 64 kHz. Each neuron was given a fit to the experimentally measured mean tuning curves using either WT or KO Gaussian parameters from the recorded ACx or IC neural data. The simulated neurons were presented with either a Go stimulus (4, 8, 16 or 32 kHz) or No-Go stimuli spaced in 1/12 octave steps up to 1 octave above and below the Go stimulus, paralleling our behavioral paradigm. Neuron activity was defined by the Gaussian tuning curve value for each presented stimulus. This

frequency-tuned activity from each neuron was then fed into a read-out layer with the assumption that the activity value for each neuron was the mean for a Poisson process. The output of each neuron in each trial was simulated by drawing a random number from this Poisson distribution with the condition that single trial activity level must surpass a threshold value (90% of the baseline firing rate) to be propagated to the readout layer. This was repeated to produce 10,000 trials for each neuron. Lastly, Bayesian decoding was used to calculate the probability that the current stimulus was a Go or No-Go based on the activity levels from this simulated neuron population. This process was repeated 15 times to illustrate model consistency and allow for statistical testing.

### Statistical analyses

Statistical analyses were performed in Python (3.10). For behavior and ABR data, two-way ANOVAs and Kruskal–Wallis Rank Sum tests were used to determine significant main effects of genotype and sound parameter (frequency/intensity), depending on normality as determined by the Shapiro–Wilks test, with Bonferroni-corrected post-hoc comparisons when applicable. Student's *t*-tests or Wilcoxon Rank Sum tests were used for single comparisons, depending on the normality. For neural data, a general linear mixed model (GLMM) analysis of variance was used to assess significant main effects of genotype and sound parameter, with individual rat ID, penetration number, and cluster number as random effects. This hierarchical approach allows for statistical comparisons at the subject level while taking into account potential within-subject variability at the unit and penetration level. All statistical values are reported as means ± standard error (SEM), unless otherwise stated. Boxplots include median (gray dash), 25th–75th percentiles (box), full range (whiskers), and outliers (below Q1 − 1.5 * IQR or above Q3 + 1.5 * IQR). Box plot dots in Fig 1 correspond to animal. Box plot dots in Figs 2 and 3 correspond to multi-unit cluster. Box plot dots in Fig 4 correspond to a full 10,000 trial run of the model for the given parameters.

### Results

#### Tone detection thresholds are unaltered in *Fmr1* KO rats

The ability to discriminate between sound frequencies is directly related to a subject's hearing thresholds, with poorer hearing thresholds associated with a decreased ability to detect small changes in frequency [41,42]. It was therefore essential to first characterize tone detection thresholds in *Fmr1* KO rats before examining any potential differences in frequency discrimination. Male *Fmr1* KO ($n = 10$) and littermate WT ($n = 9$) rats were first trained to detect tone bursts (4, 8, 16, and 32 kHz) using a Go/No-Go operant conditioning paradigm (Fig 1A). Both WT and *Fmr1* KO rats learned this task at a comparable rate (WT: 16.9 ± 1.22 days to criteria, KO: 16.6 ± 1.45 days to criteria; Wilcoxon Rank Sum: $p = 0.645$) and reached similar levels of peak performance (WT $d' = 3.70 ± 0.09$, KO $d' = 3.47 ± 0.11$; Wilcoxon Rank Sum: $p = 0.121$), consistent with our previous results [26]. In order to assess tone detection thresholds, tone stimuli were presented at near threshold intensities and psychometrics curves were constructed by plotting $d'$ as a function of sound intensity (Fig 1B). A conservative cutoff of 1.5 for $d'$ was used to calculate detection thresholds [43]. There was no significant genotype difference in psychometric functions (Fig 1B; Kruskal–Wallis: $df = 1$, $p = 0.679$) or detection thresholds (Fig 1C; Kruskal–Wallis: $df = 1$, $p = 0.974$), consistent with our previous results [26].

In addition to behaviorally measured subjective hearing thresholds, we recorded auditory brainstem responses (ABRs) in a separate group of male *Fmr1* KO ($n = 5$ rats, 10 ears) and WT ($n = 6$ rats, 10 ears) littermates (Fig 1D–1F). ABRs are a clinically used, non-invasive physiological measure of the functional status of the lower auditory pathway that allows for objective assessment of hearing thresholds (Fig 1E) [44]. We found no difference in ABR thresholds between WT and *Fmr1* KO animals across any tone frequency tested (Fig 1F; 2-way ANOVA, genotype: $df = 1$, $F = 0.867$, $p = 0.357$; genotype~frequency: $df = 40$, $F = 1.22$, $p = 0.316$), consistent with our behavioral results. These results confirm that *Fmr1* KO rats have unaltered hearing thresholds, allowing for the examination of tone discrimination performance without the confound of differences in sensation level.

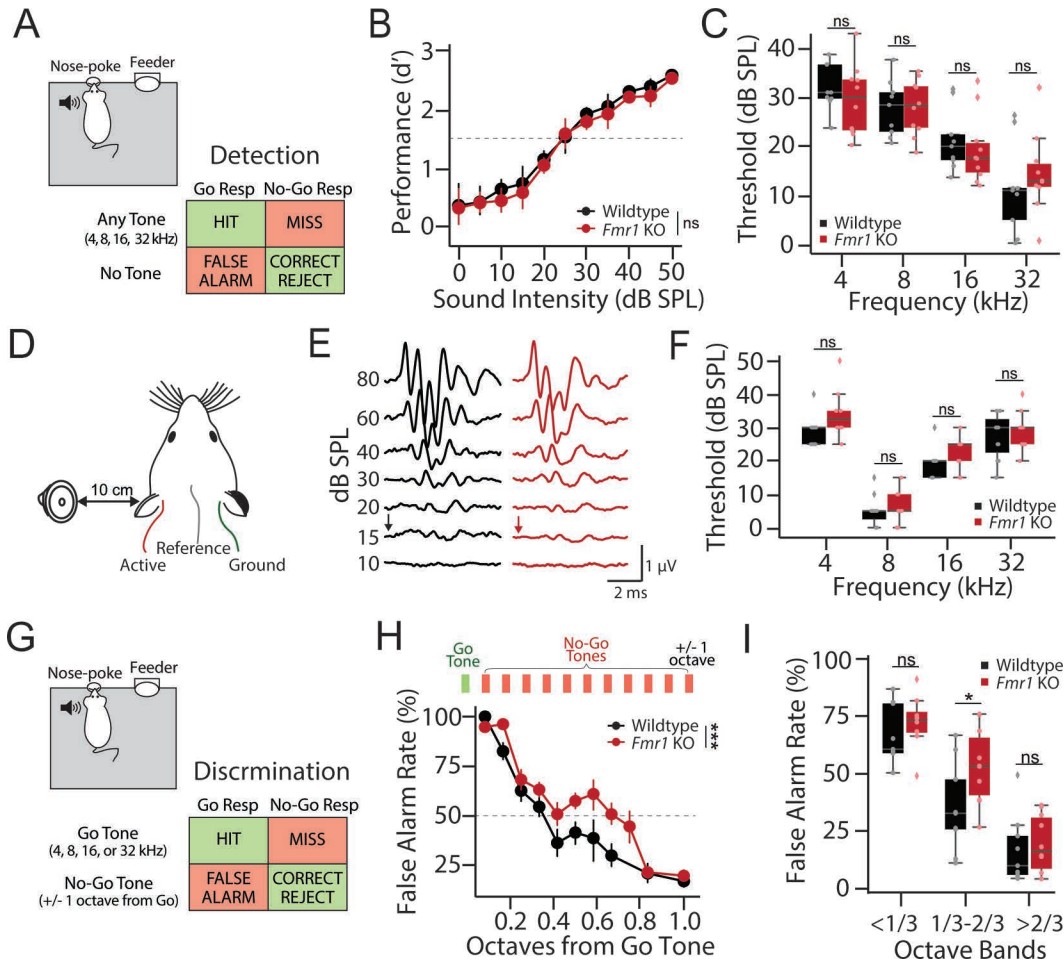

**Fig 1. Normal tone detection but impaired discrimination in *Fmr1* KO rats. (A)** Schematic of Go/No-Go operant tone detection task. 9 wildtype (black) and 10 *Fmr1* KO (red) rats were trained to report the detection of any tone burst (HIT), with failure to do so resulting in a MISS. On 30% of trials no sound was presented (catch trials). Responding on catch trials resulted in a false alarm (FA), while refraining from responding resulted in a correct rejection (CR). **(B)** Average tone detection performance across animals for all tone frequencies (4, 8, 16, and 32 kHz). Detection performance was comparable between wildtype (black) and *Fmr1* KO (red) animals. **(C)** Behavioral detection thresholds for each tone frequency using a criterion of $d' = 1.5$ (dashed line in Fig 1B). No genotype difference was observed at any test frequency. **(D)** Schematic of auditory brainstem response (ABR) recording setup. **(E)** Representative ABR waveforms from a wildtype (black) and *Fmr1* KO (red) rat. ABR threshold was defined as the lowest intensity that evoked a discernable ABR waveform (black and red arrows) **(F)** No difference in ABR thresholds between wildtype (black) and *Fmr1* KO (red) rats was observed at any sound frequency tested. **(G)** Schematic of Go/No-Go operant discrimination task. In this task, rats were trained to report the detection of a single Go-tone frequency (4, 8, 16, or 32 kHz) and inhibit their response to a No-Go tone either 1 octave above or below the Go tone. **(H)** Fine-frequency discrimination task where No-Go tone frequency was varied in 1/12 octave steps from Go tone. Top: Schematic showing the addition of multiple octave steps between the Go and No-Go tones (1/12 of an octave per step). Bottom: FA rate as a function of No-Go tone frequency in wildtype (black) and *Fmr1* KO (red) rats. **(I)** FA rates grouped by 1/3 octaves from Go showing a significant reduction in *Fmr1* KO performance only in the optimally difficult middle band, while performance is comparable at the most difficult and easy 1/3 octave bands. Box plots represent the median, 25th, and 75th percentiles. Whiskers represent the minimum and maximum values except for outliers. Boxplots dots represent individual animals. All other values are means ± SEM. *$p < 0.05$, ***$p < 0.001$, ns = not significant. Source data for panels **C, F, and I** are in S1 Data, Fig 1 sheet.

### *Fmr1* KO rats exhibit deficits in fine frequency discrimination

After ensuring similar hearing capabilities, we next moved the same group of behaviorally trained 9 WT and 10 *Fmr1* KO rats to a Go/No-Go tone discrimination task. In this phase of the task, animals were trained to respond to a single Go tone frequency (4, 8, 16, or 32 kHz, randomly assigned to genotype pairs) and inhibit their response to a No-Go tone whose

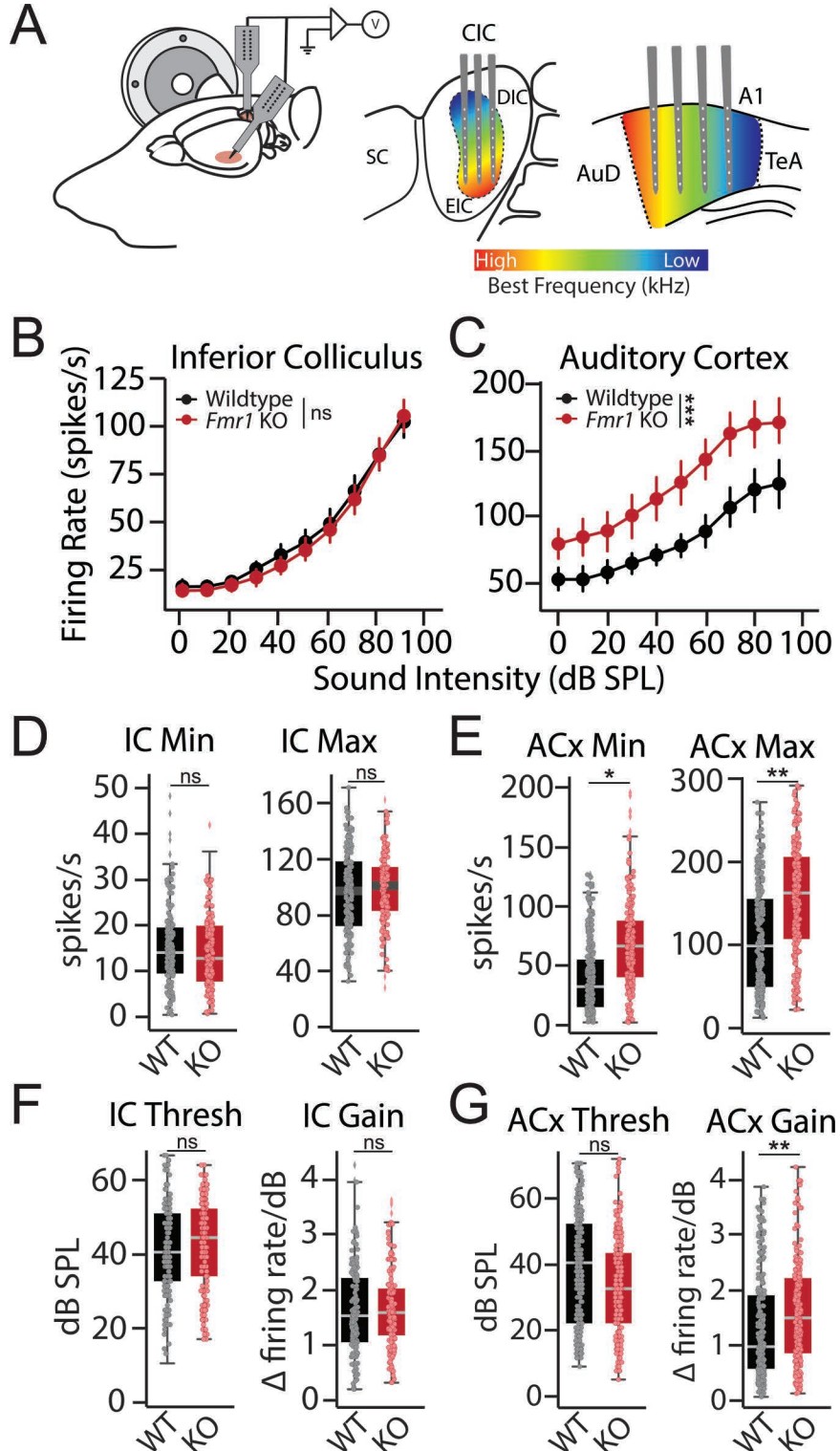

**Fig 2. Altered cortical response properties in *Fmr1* KO rats. (A)** Schematic of recording set-up. Simultaneous recordings with multichannel electrodes were made from across the tonotopic axis of contralateral auditory cortex (ACx) and inferior colliculus (IC) of 5 wildtype and 5 *Fmr1* KO rats. Recordings were made with single shank linear silicon probes that spanned the dorsal-ventral axis of the IC or ACx, with multiple penetrations being

made at different positions along the rostral-caudal axis of each area. **(B–C)** Rate level functions showing relationship between firing rate and sound intensity at characteristic frequency (CF) for each multi-unit cluster in the **(B)** IC and **(C)** ACx of wildtype (black) and *Fmr1* KO (red) rats. **(D–E)** Interpolated response minimum (Min) and maximum (Max) from **(D)** IC and **(E)** ACx response functions. **(F–G)** Interpolated response threshold (Thresh) and slope (Gain) from **(F)** IC and **(G)** ACx response functions. Box plots represent the median, 25th, and 75th percentiles. Whiskers represent the minimum and maximum values except for outliers. Boxplot dots represent individual multiunit clusters. All other values are means ± SEM. *$p < 0.05$, **$p < 0.01$, ***$p < 0.0001$, ns = not significant. Data and code underlying this figure can be found at http://doi.org/10.5281/zenodo.15559344. Source data for panels **D–G** are in S1 Data, Fig 2 sheet.

frequency was 1 octave above or below the target Go tone (Fig 1G). Both genotypes were able to reach and maintain successful criteria (WT: 14.43 ± 2.94 days to criteria, KO: 13.71 ± 2.58 days to criteria; *t* test: $t = 0.183$, $df = 18$, $p = 0.858$), with no difference in discrimination performance when tones were separated by 1 octave (S1A Fig; WT $d' = 3.39 ± 0.10$, KO $d' = 3.46 ± 0.12$; *t* test: $t = 0.427$, $df = 18$, $p = 0.675$). We next tested the limits of their discrimination by keeping the Go tone constant but bringing the frequency of the No-Go tone progressively closer to the Go tone frequency (Fig 1H). In this testing phase of the task, we found that *Fmr1* KO animals performed significantly worse than WT littermates (Fig 1H; 2-way ANOVA, genotype: $df = 1$, $F = 11.104$, ***$p = 0.001$). Importantly, other aspects of task performance were not altered in WT or *Fmr1* KO rats on discrimination testing days, including hit rate (S1B Fig), trials initiated (S1C Fig), and reaction time (S1D Fig). To further examine discrimination performance, we binned false alarms (FA) by their tonal distance from the Go frequency (Fig 1I). When the No-Go frequency was well separated from the Go tone (>2/3 octave separation), both genotypes performed well, as indicated by FA rates generally below 25%, consistent with octave discrimination performance (*t* test: $t$ value = 0.410, $p = 0.688$). Conversely, when the No-Go frequency was very close to the Go tone (<1/3 octave separation), neither genotype performed above chance levels, as indicated by the high FA rates in both WT and *Fmr1* KO animals (*t* test: $t = 0.894$, $p = 0.386$). However, for intermediate frequencies (between 1/3 and 2/3 octave separation), WT animals were significantly better at discriminating Go from No-Go tones than *Fmr1* KO littermates, as evidenced by significantly lower FA rates (*t* test: $t$ value = 2.186, *$p = 0.044$). Together, these results indicate that *Fmr1* KO rats have impaired fine feature discrimination despite having normal tone detection thresholds and learning rates.

## Divergent cortical and subcortical evoked response properties in *Fmr1* KO rats

We next sought to determine the neural mechanisms underlying impaired tone discrimination in *Fmr1* KO rats. FMRP is widely expressed in the auditory system and loss of FMRP expression has been shown to alter neuronal function across the auditory neuroaxis, including the brainstem [45–48], midbrain [49–52], and cortex [5,23]. We therefore performed simultaneous extracellular recordings of multi-unit spiking activity in the inferior colliculus (IC) and auditory cortex (ACx) of ketamine/xylazine anesthetized *Fmr1* KO ($n = 272$ ACx and 176 IC multiunit clusters from 5 rats) and WT ($n = 336$ ACx and 192 IC multiunit clusters from 5 rats) littermates. Multiunit spiking activity was assessed in response to tone bursts (50 ms) presented over a broad range of frequencies (1–64 kHz) and intensities (0–90 dB SPL) in order to examine tone-evoked activity and frequency tuning. The auditory system is tonotopically organized, with neurons tuned to different best frequencies along the dorsal-ventral axis of the IC or the rostral-caudal axis in the ACx. To account for this, multiple penetrations were made along the rostral-caudal axis of the IC and ACx, using 16 ch linear probes spanning the dorsal-ventral axis of each structure (Fig 2A). For each electrode penetration in the IC, activity was sampled across iso-frequency lamina. In the ACx, each penetration sampled activity across cortical layers from neurons roughly tuned to the same best frequency, which varied across penetrations. In this manner, we were able to assess response properties and tuning qualities at multiple levels of the auditory system across the tonotopic axis.

We constructed tone-evoked rate level functions (RLFs) from the IC (Fig 2B) and ACx (Fig 2C) by plotting the mean driven discharge rate at the unit's characteristic frequency (CF) as a function of tone intensity. No genotype difference was observed for RLFs collected from the IC (Fig 2B; GLLM: $p = 0.134$). However, we saw that tone-evoked spiking activity

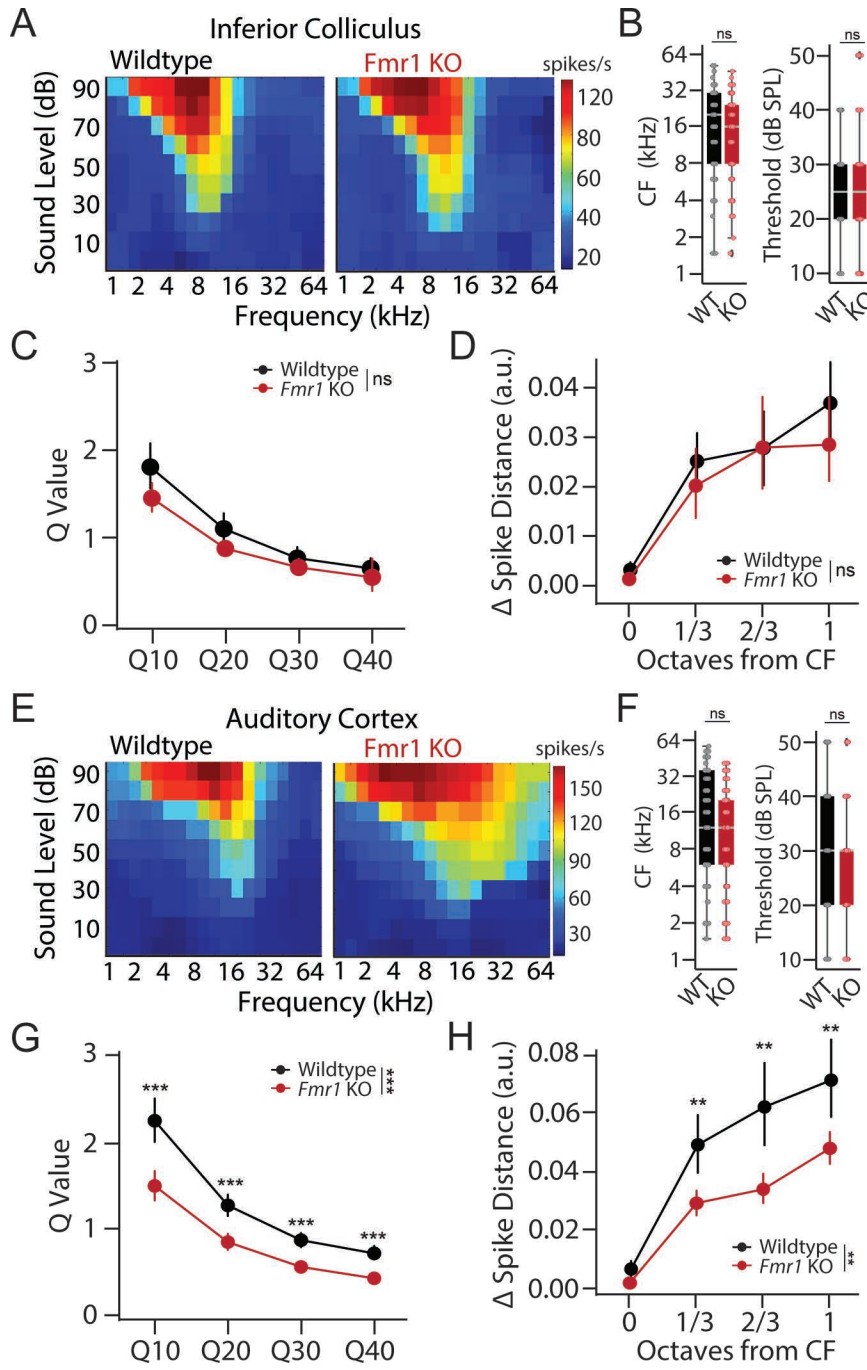

**Fig 3. Broader frequency tuning in auditory cortex of *Fmr1* KO rats despite unaltered subcortical tuning properties. (A)** Example tuning curves recorded from the inferior colliculus (IC) of a wildtype (left) and *Fmr1* KO (right) rat. Each cell represents 30 trials for a given frequency-intensity combination. **(B)** Distributions of characteristic frequency (CF, left) and minimum threshold (right) for multi-unit clusters from the IC of wildtype (black) and *Fmr1* KO (red) rats. **(C)** Q-value measure of tuning precision at 10, 20, 30, and 40 dB above response threshold for IC units. **(D)** Neural discriminability of sound frequency in IC as assessed by changes in spike train dissimilarity (Δ Spike-Distance) in response to CF and neighboring tone frequencies. **(E)** Example tuning curves recorded from the auditory cortex (ACx) of a wildtype (left) and *Fmr1* KO (right) rat. Each cell represents 30 trials for a given intensity frequency combination. **(F)** Distributions of characteristic frequency (CF, left) and minimum threshold (right) for multi-unit clusters from the ACx of WT (black) and *Fmr1* KO (red) rats. **(G)** Q-value measure of tuning precision at 10, 20, 30, and 40 dB above response threshold for ACx units. Lower Q-values in the ACx of *Fmr1* KO rats are indicative of broader tuning. **(H)** Neural discriminability of sound frequency as assessed by changes in spike

train dissimilarity (Δ Spike-Distance) in response to CF and neighboring tone frequencies. Decreased Spike-Distance in the ACx of *Fmr1* KO rats is indicative of poorer neural discriminability. Box plots represent the median, 25th, and 75th percentiles. Whiskers represent the minimum and maximum values except for outliers. Boxplots dots represent individual multiunit clusters. All other values are means±SEM. **$p < 0.01$, ***$p < 0.0001$, ns = not significant. Data and code underlying this figure can be found at http://doi.org/10.5281/zenodo.15559344. Source data for panels **B and F** are in S1 Data, Fig 3 sheet.

was significantly elevated in the ACx of *Fmr1* KO animals across intensities (Fig 2C; GLMM; genotype: *$p = 0.029$; genotype~intensity: ***$p < 0.0001$). Similar results were found when examining RLFs from all multiunit clusters in response to specific tone frequencies (S2 Fig). To further quantify response properties, RLFs for each multiunit cluster were fit with a 6 parameter gaussian function to interpolate response minimum, maximum, threshold, and gain (see "Materials and methods"). There was no significant genotype difference in any parameter for IC tone-evoked RLFs (Fig 2D and 2F; GLMM: min $p = 0.663$, max $p = 0.762$, thresh $p = 0.675$, gain $p = 0.673$). In the ACx, both response minimum (GLMM: *$p = 0.022$) and maximum (GLMM: **$p = 0.007$) were significantly elevated in *Fmr1* KO rats (Fig 2E). Cortical response gain was also significantly elevated in *Fmr1* KO rats (GLMM: **$p = 0.010$) without a change in threshold (GLMM: $p = 0.300$) (Fig 2G). These results indicate that the ACx is in a hyperexcitable state in *Fmr1* KO rats that affects both spontaneous and sound-drive spike rates, resulting in an additive shift in baseline firing rates as well as a multiplicative shift in response gain.

### Degraded cortical frequency tuning in *Fmr1* KO rats

While the above results highlight a potential cortical locus for sound hypersensitivity in *Fmr1* KO rats [26], these changes in evoked response size are not necessarily indicative of differences in frequency tuning or discrimination. To directly assess tuning properties in the IC and ACx of *Fmr1* KO rats, frequency response areas (FRAs) were examined by assessing the mean sound-evoked firing rate at each frequency-intensity combination for each multiunit cluster. Tuning properties were quantified in the following ways. First, response bandwidth was characterized by determining the *Q*-values for each multiunit cluster at multiple intensities (10–40 dB) above threshold. *Q*-values measure the sharpness of a tuning curve while accounting for differences in characteristic frequency (CF), with higher *Q*-values indicative of more precise (i.e., narrower) tuning. In addition, we employed a spike-distance-based approach to determine how well a given multiunit cluster could discriminate between different frequency tones based on the dissimilarity of the unit's spike trains (see "Materials and methods").

Representative FRAs collected from the IC of WT and *Fmr1* KO rats are shown in Fig 3A. We found a similar distribution of CF (GLLM: $p = 0.376$) and threshold (GLLM: $p = 0.245$) for multiunit clusters obtained from WT and *Fmr1* KO animals, indicating there was no bias in our sampling of responses across tonotopic regions (Fig 3B). *Q*-values became smaller in both genotypes at higher intensities (GLMM: ***$p < 0.0001$), indicative of broader tuning at higher sound levels and consistent with classic V-shaped tuning curves (Fig 3C). However, no genotype difference in *Q*-values was observed at any intensity level in the IC (GLLM: genotype: $p = 0.121$; genotype~dB: $p = 0.094$). Likewise, spike distance increased for tones that were further from the CF of each unit in both genotypes (Fig 4D, GLMM: ***$p < 0.0001$), indicating that neural discrimination improved as a function of frequency separation. However, there was no significant genotype difference in spike distance for IC units (Fig 3D, GLMM: $p = 0.365$). Together these findings show that frequency tuning is unaltered in the IC of *Fmr1* KO rats.

Representative FRAs collected from the ACx of WT and *Fmr1* KO animals are shown in Fig 3E. Again, we found a similar distribution of CF (GLLM: $p = 0.568$) and threshold (GLLM: $p = 0.190$) in both genotypes, indicating unbiased sampling (Fig 3F). However, contrary to what was found in the IC, tuning curves appeared broader in the ACx of *Fmr1* KO animals. Indeed, the *Q*-values were significantly lower in *Fmr1* KO rats across intensity levels, indicative of larger bandwidth (Fig 3G, GLMM: genotype ***$p < 0.0001$; genotype~intensity ***$p < 0.0001$). Finally, while spike-distance for ACx units once again increased as a function of frequency separation from CF in both genotypes (GLMM: ***$p < 0.0001$), the dissimilarity

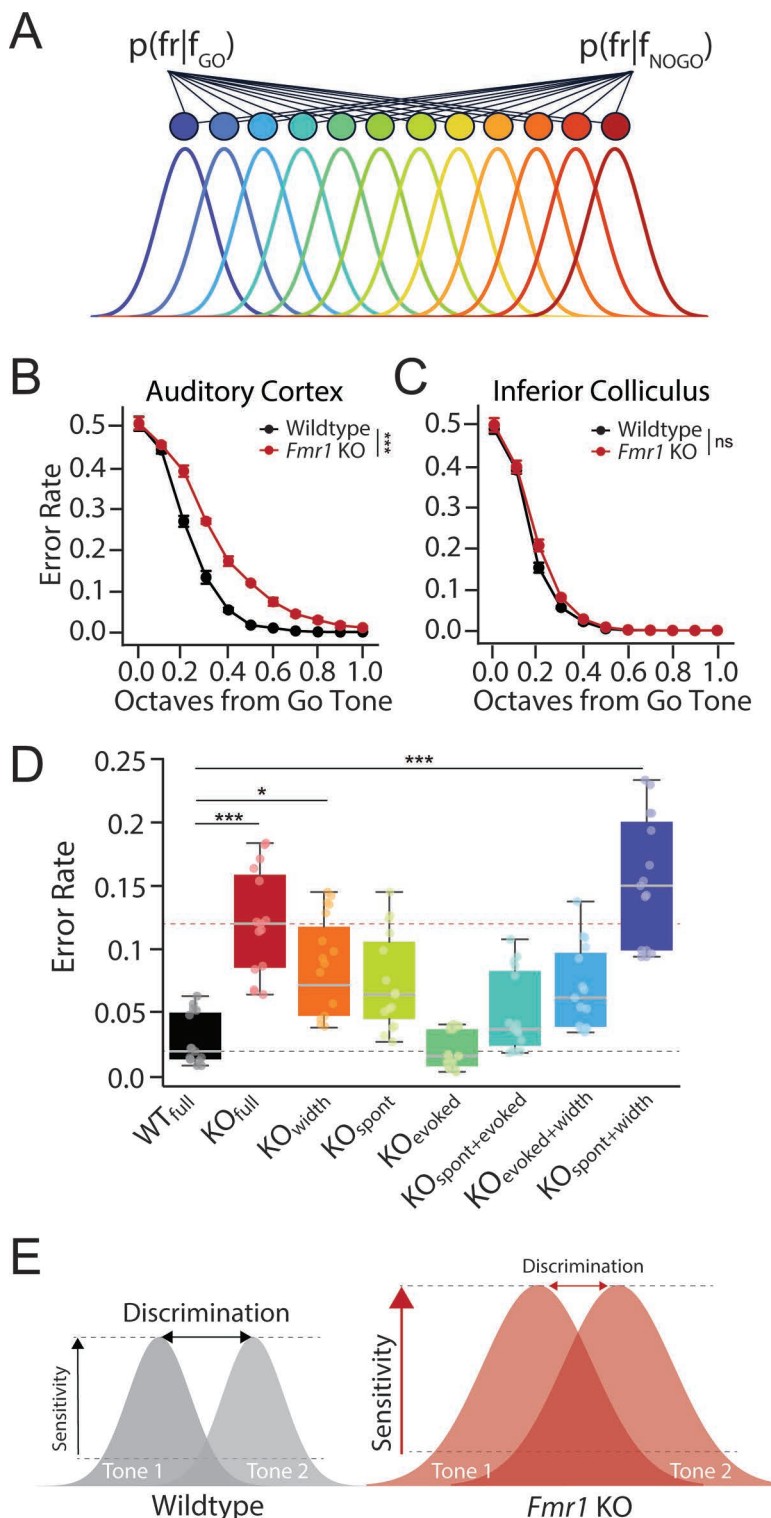

Fig 4. Modeling *Fmr1* KO discrimination deficits using a population decoder. (A) Schematic of Bayesian decoding from a simulated network of tonotopically organized neurons whose tuning parameters are derived from the population data recorded from the auditory cortex (ACx) or inferior colliculus (IC) of either WT or *Fmr1* KO rats. (B–C) Error rates from model readout layer as a function of octave distance using tuning parameters from the (B) ACx or (C) IC of WT (black) or *Fmr1* KO (red) rats. (D) Decoder performance for No-Go tones 1/3–2/3 octave from Go tone as a function of

model parameters in ACx. Error rate was determined for models using all WT (WT$_{full}$) or KO (KO$_{full}$) parameters, as well as for each unique combination of individual KO parameters for spontaneous firing rates (spont), peak sound-evoked firing rate (evoked), and tuning width (width). Changing model parameters significantly impacted decoder performance (Kruskal–Wallis Test, ***$p < 0.0001$), with KO$_{full}$ (Dunn's test, ***$p < 0.0001$), KO$_{width}$ (Dunn's test, *$p = 0.0144$) and KO$_{spont + width}$ (Dunn's test, ***$p < 0.0001$) permutations being significantly different from WT$_{full}$. Gray and pink dashed lines represent median WT$_{full}$ and KO$_{full}$ error rates. Box plots represent the median, 25th, and 75th percentiles. Whiskers represent the minimum and maximum values except for outliers. Boxplots dots represent separate model runs (10,000 repeats each). All other values are means ± SEM. *$p < 0.05$, **$p < 0.01$, ***$p < 0.0001$, ns = not significant. **(E)** Schematic summarizing results. Cortical gain enhancement in FXS may act to preserve signal-to-noise ratios and signal detection threshold at the expense of sound sensitivity and fine feature discrimination. Data and code underlying this figure can be found at http://doi.org/10.5281/zenodo.15559344. Source data for panel **D** are in S1 Data, Fig 4 sheet.

was greater for units from WT compared to *Fmr1* KO animals (GLMM: **$p = 0.002$), indicative of poorer neural discrimination in FXS (Fig 3H). Thus, despite normal subcortical tuning properties, units are more broadly tuned with poorer frequency discriminability in the ACx of *Fmr1* KO animals.

## Cortical tuning differences can account for frequency discrimination deficits in *Fmr1* KO rats

To determine if the above alterations in cortical response properties in *Fmr1* KO rats could account for the observed behavioral differences in tone discrimination, we developed a population decoder (see "Materials and methods") and compared decoder accuracy when using our experimentally measured IC or ACx tone-evoked activity from WT and *Fmr1* KO animals as input to the model (Fig 4A). To do this, we first modeled a Gaussian function using the tuning parameters at 40 dB above each unit's threshold, matching the sound levels used for our behavioral discrimination experiments. We then simulated a population of 640 neurons with characteristic frequencies spanning eight octaves (1–64 kHz), using the Gaussian models for each genotype derived from either the ACx or IC spiking data. Next, we fed this frequency-tuned activity into a layer of readout neurons, with the assumption that activity in the readout neurons follow a Poisson distribution and that only activity above a certain threshold (90% of baseline spiking rate) will be propagated to the readout layer [40]. Finally, we used a Bayesian decoder to classify between Go and No-Go tones of varying frequency difference based on the single-trial activity in the readout layer.

Error rates decreased as a function of frequency distance for both genotypes whether using neural data from the IC (2-way ANOVA: $df = 1$, ***$p < 0.0001$) or ACx (2-way ANOVA: $df = 1$, ***$p < 0.0001$). However, there was a significant genotype difference in decoder performance when using the experimentally derived mean tuning curves from the ACx as a starting point (2-way ANOVA: $df = 1$, ***$p = 0.0001$), with *Fmr1* KO ACx neural populations having significantly worse performance for intermediate Go and No-go frequency differences (approximately 1/3–2/3 octave distance), mirroring the behavioral discrimination performance (Fig 4B). This difference in genotype performance was not present in the IC tuning model (Fig 4C, 2-way ANOVA: $df = 1$, $p = 0.613$). We can thus recapitulate behavioral discrimination deficits in *Fmr1* KO rats from a population model using ACx but not IC tuning parameters, suggesting that cortical response alterations are central to frequency discrimination deficits in *Fmr1* KO rats.

We observed differences in multiple response characteristics in the ACx of *Fmr1* KO rats that could potentially contributed to impaired frequency discrimination, including elevated spontaneous firing rate, increased sound-evoked responses, and broader tuning width. We therefore systemically varied each of these parameters independently in our decoder model to determine their impact on discrimination performance (Figs 4D, S3, and S4). Tuning width had a profound effect on decoder performance, with broader tuning severely impairing discrimination. Signal-to-noise levels were also critical for frequency discrimination, with increases in baseline firing rate and/or decreases in sound-evoked firing rates degrading decoder performance (Fig S3). However, because both spontaneous and sound-evoked firing rates were increased in *Fmr1* KO rats in a similar manner, signal-to-noise ratios were maintained and these changes thus minimally impacted decoder performance (Figs 4D and S3). These results suggest that cortical gain enhancement in *Fmr1* KO animals may act to preserve signal-to-noise ratios and signal detection threshold at the expense of tuning precision and fine feature discrimination (Fig 4E).

## Discussion

In this study, we used an operant Go/No-Go tone discrimination task alongside multi-region electrophysiological recordings to characterize tone discrimination behavior and frequency tuning properties in a *Fmr1* KO rat model of FXS. We found no differences in tone detection thresholds between male WT and *Fmr1* KO animals using either subjective behavioral or objective electrophysiological measures (Fig 1A–1F). However, we did observe differences in the ability of *Fmr1* KO animals to discriminate between spectrally similar tones (Fig 1G–1I). While *Fmr1* KO rats were able to learn to discriminate tones separated by 1 octave, their performance was significantly worse than WT counterparts when challenged with tones closer in frequency (1/3–2/3 octaves). Simultaneous recordings from the ACx and IC found that subcortical response properties were relatively preserved across genotypes; however, cortical activity was altered in several ways in *Fmr1* KO rats in a manner to promote network hyperexcitability, including an additive shift in spontaneous activity, a multiplicative increase in response gain, and a broadening of frequency tuning (Figs 2 and 3). Finally, we used an experimentally informed Bayesian population decoder to demonstrate that we could recapitulate behavioral discrimination deficits from neural data recorded from the ACx but not IC, with cortical tuning width being the primary driver of performance differences (Fig 4). These results have several implications regarding the nature of auditory processing difficulties in FXS and their underlying circuit mechanisms.

Reduced sound tolerance and increased sound sensitivity (i.e., hyperacusis) are commonly observed in FXS individuals [2,17]. We previously found that *Fmr1* KO rats exhibit behavioral evidence for loudness hyperacusis in the form of faster reaction times in a sound detection task [26]. Here we show that, in addition to heightened sensitivity, *Fmr1* KO animals also exhibit degraded feature discrimination. Frequency discrimination is a fundamental aspect of sound encoding that is important for a range of auditory processing. The ability to discriminate between spectral features is critical for the identification and localization of a sound source [53–56]. Degraded frequency discrimination can effect speech perception under all listening conditions, but would most severely impact speech-in-noise perception [54,57]. Thus, impaired frequency discrimination in FXS is likely to contribute to difficulties in speech comprehension and sound segregation in noisy environments [29]. While hypersensitivity has rightfully been a major focus in regards to auditory processing difficulties in FXS [2], these results highlight that, in addition to increased loudness sensitivity, degraded feature encoding is also likely to be a major contributor to reduced sound tolerance and sensory overload often observed in FXS individuals.

What are the neural mechanisms driving aberrant auditory discrimination in FXS? Here, we demonstrated that *Fmr1* KO rats exhibit broader tuning in the ACx despite finding no difference in IC response bandwidth. We could also recapitulate behavioral discrimination deficits in our population decoder when using tuning data from the ACx but not IC of *Fmr1* KO rats, and our model indicated that cortical tuning width was the main factor driving discrimination differences. Taken together, our results implicate cortical frequency tuning as being key to frequency discrimination impairments in *Fmr1* KO rats. This is consistent with previous work in *Fmr1* KO mice demonstrating that degraded orientation tuning in primary visual cortex underlies visual discrimination impairments [11] and that deficits in whisker-guided behavior are associated with blurred somatotopic maps [24]. Thus, degraded sensory tuning appears to be a conserved coding deficit observed across multiple sensory modalities in *Fmr1* KO rodents. Altered sensory tuning is not unique to FXS either, as impaired sensory discrimination has been reported in the ASD population more generally [58–60] and many genetic and environmental models of ASD also present with abnormal tuning and/or distorted sensory maps [61–65]. Interestingly, ASD models characterized by either hyper- or hypo-sensitivity can both exhibit degraded sensory tuning, suggesting this may be a fundamental mechanism for disrupted sensory processing in ASD [30].

In addition to altered tuning, we also observed elevated spontaneous and tone-evoked activity in the ACx of *Fmr1* KO rats. These findings are consistent with previous EEG studies in FXS humans and *Fmr1* KO mice demonstrating increased amplitude of the N1 wave in response to tones and altered resting-state neural oscillations [5,6,9,10,66]. Changes in cortical evoked response magnitude may account for heightened perceptual sensitivity in FXS. Indeed, we have previously shown that changes in ACx evoked response size are well-correlated with

reaction time changes on a subject-to-subject basis in drug and noise-induced hearing loss models of hyperacusis [33,36]. In addition to heightened sensitivity, we also previously observed abnormal perceptual integration of stimulus bandwidth in *Fmr1* KO rats [26], and this increased sensitivity to changes in bandwidth could be related to the altered cortical frequency tuning. If neurons are more broadly tuned in the ACx, then a larger population will be recruited with increases in stimulus bandwidth, leading broader spectrum sounds to be perceived as more intense. In this manner, abnormal frequency tuning and spectral integration could also contribute to heightened sound sensitivity and hyperacusis in FXS. Conversely, changes to spontaneous and sound-evoked activity may also influence discrimination behavior [40], as our population model indicated that changes to signal-to-noise ratios also greatly impacted decoder performance. However, the increase in both spontaneous and sound-evoked activity observed in the ACx of *Fmr1* KO rats acted to preserve signal-to-noise ratios and thus minimally impacted decoder performance. These results may indicate that there are distinct neural mechanisms underlying auditory sensitivity and discrimination alterations in FXS. However, it is possible that these diverse auditory phenotypes emerge from a common underlying cause.

Sensory systems must be sensitive enough to detect faint signals but also selective enough to differentiate between stimuli with similar features. Previous work has shown that gain control mechanisms in the ACx allow for the dynamic switching between feature detection and discrimination, with cortical gain increases biasing sound processing toward hypersensitivity and improved signal detection, while gain reductions dampen excitability and enhance frequency discrimination [67]. Here we find evidence for increased response gain in the cortex of *Fmr1* KO animals, despite normal subcortical response properties. It is thus possible that gain control mechanisms are fundamentally disrupted in FXS, with a maladaptive increase in cortical gain predisposing *Fmr1* KO animals to heightened perceptual sensitivity at the expense of fine feature discrimination (Fig 4E). Importantly, both of these changes are likely to contribute to reduced tolerance and atypical sound processing, suggesting that a single mechanism has the potential to account for diverse auditory symptoms in FXS. Future work must therefore continue to dissect the cellular and circuit mechanisms underlying cortical hyperexcitability in FXS in order to develop novel therapies for these often-debilitating sensory symptoms.

## Supporting information

**S1 Fig.  Tone discrimination performance in wildtype and *Fmr1* KO rats. (A)** Hit and false alarm (FA) rates over training sessions for 9 wildtype (black) and 10 *Fmr1* KO (red) rats. No significant difference in hit rate (2-way ANOVA, genotype: $df = 1$, $F = 1.031$, $p = 0.3284$) or FA rate (2-way ANOVA, genotype: $df = 1$, $F = 0.0004$, $p = 0.9828$) was observed between wildtype and *Fmr1* KO animals, demonstrating that both genotypes can learn and perform octave discrimination at comparable levels. **(B)** Comparison of average hit rate on the last two days of octave training (Train) and discrimination testing days (Test). There was no significant effect of task condition (Kruskal–Wallis: $df = 1$, $p = 0.891$) or genotype (Kruskal–Wallis: $df = 1$, $p = 0.356$) on hit rate. **(C)** Comparison of average trial count on the last two days of octave training (Train) and discrimination testing days (Test). There was no significant effect of task condition (2-way ANOVA: $df = 1$, $F = 1.908$, $p = 0.159$) or genotype (2-way ANOVA: $df = 1$, $F = 0.031$, $p = 0.860$) on trial count and no significant genotype~condition interaction (2-way ANOVA: $df = 2$, $F = 0.198$, $p = 0.821$). **(D)** Comparison of average reaction time on the last two days of octave training (Train) and discrimination testing days (Test). There was no significant effect of task condition (2-way ANOVA: $df = 1$, $F = 0.506$, $p = 0.679$) or genotype (2-way ANOVA: $df = 1$, $F = 0.008$, $p = 0.930$) on reaction time and no significant genotype~condition interaction (2-way ANOVA: $df = 2$, $F = 0.069$, $p = 0.933$). Box plots represent the median, 25th, and 75th percentiles. Whiskers represent the minimum and maximum values except for outliers. Boxplots dots represent individual animals. All other values are means ± SEM. ns = not significant. Source data for panels **B–D** are in S1 Data, S1 sheet.
(PDF)

**S2 Fig. Tone specific rate-levels functions from the inferior colliculus and auditory cortex of wildtype and *Fmr1* KO rats. (A)** Multi-unit spiking activity recorded from the inferior colliculus of wildtype (WT, black) and *Fmr1* KO (red) littermates in response to individual frequencies (4, 8, 16, and 30 kHz) across intensities (0–90 dB SPL, 10 dB steps). No significant genotype difference was observed at any frequency (GLLM: 4 kHz: $p = 0.402$, 8 kHz: $p = 0.577$, 16 kHz: $p = 0.248$, 32 kHz: $p = 0.843$). **(B)** Multi-unit spiking activity recorded from the auditory cortex of wildtype (WT, black) and *Fmr1* KO (red) littermates in response to individual frequencies (4, 8, 16, and 30 kHz) across intensities (0–90 dB SPL, 10 dB steps). Significant genotype differences were observed at each individual frequency (GLMM: 4 kHz: *$p = 0.012$, 8 kHz: *$p = 0.014$, 16 kHz: **$p = 0.001$, 32 kHz: *$p = 0.024$). All values are means ± SEM. Data and code underlying this figure can be found at http://doi.org/10.5281/zenodo.15559344.
(PDF)

**S3 Fig. Effect of varying auditory cortex model parameters on decoder accuracy. (A)** Heatmaps for error rate as a function of systematically varying auditory cortex (ACx) tuning width for the wildtype (WT, left) and *Fmr1* KO (right) models. Both WT and KO models appear to be equally sensitive to increased tuning bandwidth. **(B)** Heatmaps for error rate as a function of systematically varying ACx noise parameter (reflecting spontaneous firing rates) for the WT (left) and *Fmr1* KO (right) models. The KO model appears to be more sensitive to increases in spontaneous firing rates compared to WT. **(C)** Heatmaps for error rate as a function of systematically varying ACx amplitude parameter (reflecting peak sound-evoked firing rates) for the WT (left) and *Fmr1* KO (right) models. Both models are sensitive to reduction in amplitude, but KO model has consistently poorer accuracy at all amplitude levels.
(PDF)

**S4 Fig. Effect of varying inferior colliculus model parameters on decoder accuracy.** Decoder performance for No-Go tones 1/3–2/3 octave from Go tone as a function of model parameters using physiological data from inferior colliculus (IC) of wildtype (WT) and *Fmr1* KO (KO) animals. Error rate was determined for models using all WT (WT$_{full}$) or KO (KO$_{full}$) parameters, as well as for each unique combination of individual KO parameters for spontaneous firing rates (spont), peak sound-evoked firing rate (evoked), and tuning width (width). Gray and pink dashed lines represent median WT$_{full}$ and KO$_{full}$ error rates. While there was a significant effect of model parameter on decoder performance (Kruskal–Wallis Test, *$p < 0.0179$), there was no difference in performance between any KO model permutation compared to WT$_{full}$ (Dunn's test, $p = 1.0$). Box plots represent the median, 25th, and 75th percentiles. Whiskers represent the minimum and maximum values except for outliers. Boxplots dots represent separate model runs (10,000 repeats each). Source data for this figure are in S1 Data, S4 sheet.
(PDF)

**S1 Data. An excel file containing the individual data values that create the summary data figures in the manuscript.**
(XLSX)

## Acknowledgments

We would like to thank Dr. Richard Salvi for the generous gift of the *Fmr1^tm1sage^* rats. We thank the Auerbach laboratory members for their valuable feedback on the manuscript.

## Author contributions

**Conceptualization:** Benjamin D. Auerbach.

**Data curation:** D. Walker Gauthier, Noelle James, Benjamin D. Auerbach.

**Formal analysis:** D. Walker Gauthier, Noelle James, Benjamin D. Auerbach.

**Funding acquisition:** Benjamin D. Auerbach.

**Project administration:** Benjamin D. Auerbach.

**Supervision:** Benjamin D. Auerbach.

**Writing – original draft:** D. Walker Gauthier, Benjamin D. Auerbach.

**Writing – review & editing:** D. Walker Gauthier, Noelle James, Benjamin D. Auerbach.

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
