## [Editor Report · Decision Letter 0]

Dear Ben, 

Thank you for submitting your manuscript entitled "Altered auditory feature discrimination in a rat model of Fragile X Syndrome" for consideration as a Research Article by PLOS Biology.

Your manuscript has now been evaluated by the PLOS Biology editorial staff and I am writing to let you know that we would like to send your submission out for external peer review. However, I am not convinced, given the level of mechanistic insight, that the manuscript would work as full Research Article for us. I think it would work better as a Short Report. This would mean to reduce the number of figures to four, but I think this would be possible, so I'd encourage to submit your manuscript as Short Report. You do not need to reduce the number of figures now already. This is something we can work on if we were to invite a revision.

Once your full submission is complete, your paper will undergo a series of checks in preparation for peer review. After your manuscript has passed the checks it will be sent out for review. To provide the metadata for your submission, please Login to Editorial Manager (https://www.editorialmanager.com/pbiology) within two working days, i.e. by Feb 23 2025 11:59PM.

Kind regards,

Christian

Christian Schnell, PhD

Senior Editor

PLOS Biology

cschnell@plos.org

---

## [Decision Letter · Decision Letter 1]

Dear Dominik,

Thank you for your patience while your manuscript "Altered auditory feature discrimination in a rat model of Fragile X Syndrome" went through peer-review at PLOS Biology. Your manuscript has now been evaluated by the PLOS Biology editors, an Academic Editor with relevant expertise, and by several independent reviewers.

In light of the reviews, which you will find at the end of this email, we are pleased to offer you the opportunity to address the concerns from the reviewers in a revision that we anticipate should not take you very long. We will then assess your revised manuscript and your response to the reviewers' comments with our Academic Editor aiming to avoid further rounds of peer-review, although we might need to consult with the reviewers, depending on the nature of the revisions.

As discussed previously, our Short Reports format allows only four figures. To achieve this, it would probably work best to combine figures 2 and 3, and figures 4 and 5, but we are open to other suggestions, as long as you don't have more than four figures.

**IMPORTANT - SUBMITTING YOUR REVISION**

*Resubmission Checklist*

*Published Peer Review*

*PLOS Data Policy*

*Blot and Gel Data Policy*

Sincerely,

Christian

Christian Schnell, PhD

Senior Editor

PLOS Biology

cschnell@plos.org

REVIEWS:

Reviewer #1: In this timely study, authors perform a thorough analysis of the electrophysiological and behavioral alterations in processing auditory information using rat model of FXS. The study is novel and report impaired auditory frequency resolution in Fmr1 KO rats using behavioral Go/no-Go task. In addition, authors perform in vivo electrophysiological recording and show that the behavioral alterations correlate with neuronal hyperactivity and broader frequency tuning in the auditory cortex but not IC of Fmr1 KO rat. Authors also report that the impaired behavioral and electrophysiological discriminations are observed between spectrally similar tones that are closer in frequency, while no changes in overall tone detection are observed in Fmr1 KO rats. These are very interesting observations that are consistent with findings reported in Fmr1 KO mice and humans with FXS. The studies also suggest that gain enhancement in the cortex may act to preserve signal-to-noise ratios and signal detection threshold at the expense of tuning precision and fine discrimination. The manuscript is well written and the reviewer has no concerns related to the design, presentation, analysis or interpretation of the study. I believe that the manuscript is acceptable for publication in its present form.

Reviewer #2: This manuscript describes a study in male Fmr1 KO rats that used behavioral testing, in vivo electrophysiology, and a computational approach to analyze differences in sound frequency discrimination in KO rats versus WT littermates. The study found that Fmr1 KO rats have a deficit in sound frequency discrimination that is associated with hyperexcitability in the auditory cortex, but not in the IC, and broader frequency tuning curves. They built a decoder to verify that these differences in neuronal responses can account for the observed behavioral effect.

The study seems to be sound and well-performed. The data is presented in a concise way and the manuscript is very well written. I have no major concerns, just a few comments to consider: 

1. It would have been really interesting to see to what extent these findings extend to female rats. While I can see that rationale for using only male rats, it would really be valuable to follow up with females.

2. The results parallel some findings in other genetic rodent autism models - maybe worthwhile to briefly discuss?

Minor: Fig. 5D is missing a title for the y-axis. 

Reviewer #3: PLOS Biology 

Altered auditory feature discrimination in a rat model of Fragile X Syndrome 

Gauthier et al. investigate tone discrimination in Fragile X syndrome knockout rats (Fmr1 KO). In their previous study (Auerbach et al, 2021), they demonstrated that these KO rats exhibit auditory hypersensitivity, characterized by faster reaction times in a go/no-go sound detection task. In the current study, they further show that Fmr1 KO rats have impaired tone discrimination abilities. Electrophysiological recordings in anesthetized rats reveal that frequency tuning in the inferior colliculus remains unchanged, whereas the auditory cortex exhibits broader tuning in KO rats. While these findings are interesting, the data analysis lacks some key details that are necessary to interpret the results. These are listed below.

Comments:

In Figure 1, the results from the go/no-go detection task and ABR recordings indicate that KO rats have similar threshold detection across tested frequencies compared to wildtype rats. However, there appears to be a discrepancy between Figures 1C and 1F. Behaviorally, higher frequencies (16, 32 kHz) are detected more easily than lower frequencies (4, 8 kHz), yet ABR data in Figure 1F suggest that low (4 kHz) and high (32 kHz) frequencies have similar thresholds. Could the authors clarify this inconsistency?

Line 331: Fmr1 KO (n = 5 rats, 10 ears) and WT (n = 6 rats, 10 ears).

If there are 6 WT rats, there should be 12 ears. Is this a mistake or some ears were excluded from the analysis for some reason?

The figure legends are missing key information, including:

a) What each error bar represents

b) What each dot in the box plots correspond to.

Figure 4A and 5A panels are missing colorbar values. Should they be spikes/sec?

Line 448: The text refers to "Figure 4E," but this panel does not seem to be included. 

Authors state that the neuronal activity was sampled across iso-frequency lamina in the IC and cortical layers in the ACx. Based on Figure 3A, it would be expected that the position of the electrode shank could be determined based on best frequency, but no data (functional or histological) is provided to support this claim.

In addition to my previous comment, in Figure 3A, it seems that recordings were done in multiple subregions of the IC, but this is not clarified in the manuscript. Was the electrophysiological recording performed in the central inferior colliculus, or did it span through multiple subregions? If so, could the authors show that there are no functional differences between these subregions, particularly considering the high variability observed in Figure 3D and F?

In Figure 2C and D, only the false alarm rate is shown. It would be interesting to know if the hit rate also changes when multiple no-go frequencies are introduced. Additionally, since reaction times were extensively studied in a previous publication (tone detection task; Auerbach et al, 2021), could the authors provide the reaction times for the hit trials in the discrimination task as well?

For Figure 2, are KO rats generally unable to discriminate similar tones, or is this effect frequency-dependent? If so, is it consistent across multiple sessions?

In Figure 3, there is a big variability in the data for both wildtype and KO groups within the auditory cortex and inferior colliculus. Is this variability due to between-rat or within-rat differences?

Although the authors report a significant difference between WT and KO groups in the auditory cortex, the distributions of the two groups are highly overlapping. Could the authors discuss whether this significance is meaningful given the overlap?

Figures 4B and 5B lack statistical tests to compare WT and KO groups. What do these results suggest?

In Figure 6D, what do the individual dots represent? What are the two dashed lines?

This figure only shows decoding accuracy results for the auditory cortex, but it would be helpful to see similar analyses for the inferior colliculus. Additionally, to show differences for the auditory cortex in KO rats, a statistical test is needed.

---

## [Decision Letter · Decision Letter 2]

Dear Ben,

Thank you for your patience while we considered your revised manuscript "Altered auditory feature discrimination in a rat model of Fragile X Syndrome" for publication as a Short Reports at PLOS Biology. This revised version of your manuscript has been evaluated by the PLOS Biology editors, the Academic Editor and one of the original reviewers.

Based on the reviews and on our Academic Editor's assessment of your revision, we are likely to accept this manuscript for publication, provided you satisfactorily address the following data and other policy-related requests:

* Please modify the last sentence of the abstract. We think that the current study is too far removed from potential medical applications and want to avoid raising false hopes. We think you can simply delete the second half of the sentence (after the comma).

* Please add the links to the funding agencies in the Financial Disclosure statement in the manuscript details.

* DATA POLICY:

Regardless of the method selected, please ensure that you provide the individual numerical values that underlie the summary data displayed in the following figure panels as they are essential for readers to assess your analysis and to reproduce it: 1CFI, 2DEFG, 3BF, 4D, S1BCD and S4

* CODE POLICY

We expect to receive your revised manuscript within two weeks. 

*Published Peer Review History*

*Press*

Sincerely,

Christian

Christian Schnell, PhD

Senior Editor

cschnell@plos.org

PLOS Biology

Reviewer remarks:

Reviewer #2: I think that the authors have thoroughly addressed the concerns of reviewer #2 and reviewer #3. I have no further comments and congratulate the authors to this study.

---

## [Editor Report · Decision Letter 3]

Dear Ben,

Thank you for the submission of your revised Short Reports "Altered auditory feature discrimination in a rat model of Fragile X Syndrome" for publication in PLOS Biology. On behalf of my colleagues and the Academic Editor, Jennifer Bizley, I am pleased to say that we can in principle accept your manuscript for publication, provided you address any remaining formatting and reporting issues. These will be detailed in an email you should receive within 2-3 business days from our colleagues in the journal operations team; no action is required from you until then. Please note that we will not be able to formally accept your manuscript and schedule it for publication until you have completed any requested changes.

PRESS

Sincerely, 

Christian

Christian Schnell, PhD

Senior Editor

PLOS Biology

cschnell@plos.org